# A Case of Acquired Reactive Perforating Dermatosis with Complete Resolution of Eruptions on Upper and Lower Limbs During the Treatment of Diabetes Mellitus and Peripheral Artery Disease

**DOI:** 10.3390/medicina61010036

**Published:** 2024-12-29

**Authors:** Yoshihito Mima, Tsutomu Ohtsuka, Ippei Ebato, Ryosuke Nishie, Satoshi Uesugi, Makoto Sumi, Yoshimasa Nakazato, Yuta Norimatsu

**Affiliations:** 1Department of Dermatology, Tokyo Metropolitan Police Hospital, Tokyo 164-8541, Japan; 2Department of Dermatology, International University of Health and Welfare Hospital, Tochigi 329-2763, Japanippei.eba0717@gmail.com (I.E.); 3Department of Vascular Surgery, International University of Health and Welfare Hospital, Tochigi 329-2763, Japan; nishie@iuhw.ac.jp (R.N.); r1407us@iuhw.ac.jp (S.U.); sumi@iuhw.ac.jp (M.S.); 4Department of Diagnostic Pathology, International University of Health and Welfare Hospital, Tochigi 329-2763, Japan; nakazato@iuhw.ac.jp; 5Department of Dermatology, International University of Health and Welfare Narita Hospital, Chiba 286-8520, Japan; norimanorima@gmail.com

**Keywords:** acquired reactive perforating dermatosis, diabetic mellitus, peripheral artery disease, microangiopathy, ischemia

## Abstract

Acquired reactive perforating dermatosis (ARPD) is characterized by its onset after the age of 18 years, umbilicated papules or nodules with a central keratotic plug, and the presence of necrotic collagen tissue within an epithelial crater. ARPD is strongly associated with systemic diseases such as diabetes mellitus (DM) and chronic renal failure, which may contribute to ARPD through factors including microcirculatory disturbances and the deposition of metabolic byproducts, including advanced glycation end-products and calcium. Here, we report a case of ARPD that improved following DM treatment and catheter-based interventions for peripheral artery disease (PAD). The eruptions on the upper limbs significantly improved with DM management. On the other hand, lesions on the lower limbs showed marked improvement after the enhancement in arterial blood flow due to catheter surgeries, along with DM treatment. Although a few reports of ARPD improving with DM management exist, our case underscores the importance of adequate DM control in ARPD management. The inability to perform the biopsy of the lesions on the lower limbs is our limitation; however, these lesions, similar to those on the upper limbs, presented with a central keratotic plug and re-epithelialized without forming ulcers or erosions, suggesting they were also related to ARPD. To date, there has been little discussion on the relationship between blood flow impairment in major vessels and ARPD. However, hypertension and venous circulatory dysfunctions are considered to lead to ARPD, raising the possibility that PAD-induced microvascular disturbances might have facilitated lesion formation in the present case. Further accumulation of cases and research is needed to clarify the relationship between blood flow impairment in major vessels and ARPD.

## 1. Introduction

Perforating dermatoses are heterogeneous skin disorders characterized by transepidermal elimination of dermal components, particularly collagen and elastic fibers [1]. Four classical forms of primary perforating dermatosis have been documented, where the transepidermal elimination mechanism represents the hallmark of the disease as follows: elastosis perforans serpiginosa, Kyrle’s disease and perforating folliculitis, and acquired reactive perforating dermatosis (ARPD) [2]. Elastosis perforans serpiginosum presents with the elimination of elastic fibers. Kyrle’s disease presents with transepidermal elimination of abnormal keratin. Perforating folliculitis presents with transepidermal elimination of the follicle. ARPD presents with transepidermal elimination of collagen fibers [2].

ARPD diagnosis is based on histopathological analysis, age of onset, and characteristic skin lesions [3]. Faver et al. stated the diagnostic criteria for ARPD, including onset after the age of 18 years, umbilicated papules or nodules with a central keratotic plug, and the presence of necrotic collagen tissue within an epithelial crater [3].

Although the precise pathogenesis of ARPD remains unclear, various factors, including chronic venous insufficiency [4], collagen degeneration [5], increased glycation end-products in skin lesions [6], and overexpression of transforming growth factor-beta 3 (TGF-β3) in extracellular matrix remodeling and wound healing [7] have been reported to possibly contribute to ARPD development. Additionally, th2-mediated cytokines, such as interleukin (IL)-4 and IL-13, are also considered to play a role, similar to mechanisms observed in atopic dermatitis [8].

ARPD has been reported to be connected mainly with diabetes mellitus (DM) and chronic renal failure (CRF) [4]. Currently, various systemic conditions, such as chronic venous insufficiency, primary sclerosing cholangitis, thyroid disease, dermatomyositis, and pulmonary fibrosis, have also been considered to lead to ARPD occurrence [4,9,10,11,12]. ARPC has also been regarded as a paraneoplastic symptom in hematologic malignancies, such as Hodgkin’s lymphoma, and solid tumors, including hepatocellular carcinoma [13,14].

The mechanism by which DM contributes to ARPD may involve microcirculatory disturbances leading to dermal collagen degeneration and inflammation caused by the accumulation of glycation end-products [6,15,16]. In renal failure, abnormal metabolic deposits in the dermis, such as calcium, along with circulatory impairment, are hypothesized contributors [16]. Elevated immunoreactivity of TGF-β3, matrix metalloproteinase-1, and the tissue inhibitor of metalloproteinases-1 in ARPD lesions suggests a potential link between ARPD and CRF [7].

Drug-induced ARPD has been reported with agents such as clopidogrel [17], erlotinib [18], and chemotherapy regimens, including sorafenib and sirolimus [19,20]. Other possible triggers include insect bites [21], scabies [22], trauma [23], and pregnancy [24].

The treatment options for ARPD include topical glucocorticoids and calcineurin inhibitors and oral medications such as doxycycline and allopurinol [25,26]. Oral antihistamines and phototherapy can be effective in disrupting the itch–scratch cycle, thereby alleviating pruritus [25,27]. Allopurinol may contribute to the improvement of ARPD by inhibiting xanthine oxidase, reducing free radicals that damage collagen, and preventing collagen cross-linking induced by advanced glycation end-products [28,29].

Recently, biologic agents, such as dupilumab and nemolizumab, as well as Janus Kinase (JAK) inhibitors like baricitinib and upadacitinib, have shown promise in the treatment of ARPD [8,30,31,32]. In ARPD lesions, elevated expression of IL-4 and IL-13, along with marked infiltration of Th2 cells, has been observed, suggesting that dupilumab may be an effective treatment option for ARPD [8,33,34]. IL-31 has been associated with fibrosis in both the skin and lungs, and increased IL-31 expression has been noted in ARPD and other fibrotic conditions such as pulmonary fibrosis [30,35,36]. Thus, nemolizumab may prove to be effective in treating ARPD by inhibiting IL-31 signals [30]. Additionally, JAK inhibitors such as baricitinib [31] and upadacitinib [32] may offer therapeutic benefit in ARPD by suppressing Th2-mediated immune responses and cytokine activity [31,32].

Beyond pharmacological treatments, managing underlying conditions, including DM, CRF, and malignancies, might lead to improved ARPC outcomes [37]. Case reports have detailed the resolution of ARPD lesions with moisturizers alone following surgical excision of malignancies, such as hepatocellular carcinoma [38,39], while one case of ARPD has shown remarkable regression with intensified DM management and moisturizer use over 4 months [40].

Here, we report a case of ARPD which shows significant improvement following the treatment of DM and peripheral artery disease (PAD).

## 2. Case Presentation

A 73-year-old Japanese woman had been diagnosed with type 2 DM over a decade ago; however, due to only mild elevations in her hemoglobin A1c (HbA1c) level, she had been managed without any medications. Approximately two years ago, she developed multiple papules on both her upper and lower extremities around the same time. These lesions proved refractory to treatment with betamethasone dipropionate ointment and oral olopatadine. Physical examination revealed multiple itchy red papules with central sebaceous plugs on her extremities, painful swelling in her lower left leg, and a black scab on her left index finger (Figure 1a,b).

Histopathological examination of upper extremity lesions revealed a cup-shaped defect filled with a mixture of inflammatory cells, degenerated collagen fibers, and necrotic material. Beneath this plug, an infiltration of predominantly lymphocytic inflammatory cells was observed (Figure 2a). Additionally, Elastica van Gieson staining revealed degenerated fiber bundles being extruded vertically from the superficial dermis (Figure 2b).

An additional biopsy of the lower limbs was also suggested; however, she refused this due to severe pain in the lower limbs. Eliminating degenerated fibers, cutaneous symptoms, and the occurrence period led to the diagnosis of ARPD [1,2,3]. The perforating dermatosis was classified as severe (16.4) [1,2,3]. Laboratory examination revealed elevated HbA1c (10.8%) and glucose (382 mg/dL) and normal creatinine level (0.75 mg/dL). Contrast computed tomography revealed arterial narrowing and ischemia of the left common iliac artery (CIA) and superficial femoral artery (SFA), indicating PAD (Figure 3).

In order to control postprandial hyperglycemia and enhance insulin secretion, oral voglibose and oral sitagliptin phosphate hydrate were initiated. Due to the urgent need for rapid blood glucose reduction, insulin injections were also administered. Prior to endovascular catheter therapy, alprostadil injections, a vasodilator, were introduced to improve blood flow. After 2 months of DM treatment, with a modest improvement in hemoglobin A1c levels from 10.8% to 7.5%, the central crusts of the papules and erythema on the upper extremities tended to fade over time and become flatly pigmentated without forming ulcers or erosions. However, lesions on the lower limb remained unchanged (Figure 4a,b).

To restore the blood flow in the lower leg, left CIA stenting (Figure 5a) and SFA balloon angioplasty (Figure 5b) were performed. The blood flow in left SFA dramatically improved through these catheter surgeries (Figure 5c).

The eruptions closer to the center of the lower leg showed complete regression 3 months post-endovascular therapy; however, the erythematous area with an internal large crust still retained the crust (Figure 6a). Eruptions on the lower limbs became flatly pigmentated without central crusts 6 months post-endovascular therapy (Figure 6b). The central crust of the erythema on the lower limbs, similar to that on the upper limbs, regressed over time without presenting ulcers or erosions. The blood glucose levels remained well controlled, with HbA1c stabilizing at 6.7%. The timeline and progression of the onset of DM, PAD, and skin lesions on the upper and lower limbs are shown in Figure 7.

## 3. Discussion

The eruptions on the lower limbs took longer to improve, compared to those on the upper limbs. However, the lesions on the lower limbs showed marked improvement after the enhancement in arterial blood flow due to catheter surgeries, along with DM treatment. Differential diagnoses of the skin rash on the lower limbs included other types of perforating dermatoses, vasculitis such as nodular polyarteritis, thrombogenic conditions like cryoglobulinemia, metabolic disorders such as amyloidosis, autoimmune diseases like bullous pemphigoid, granulomatous diseases such as sarcoidosis and granuloma annulare, porokeratosis, and malignant tumors such as squamous cell carcinoma. However, the cutaneous histopathological findings showed no evidence of neutrophilic inflammation around vessels, deposition of metabolic byproducts, malignancy, or granuloma formation. Therefore, all differential diagnoses other than ARPD were ruled out. Although we also considered the possibility that the lesions on the lower extremity could be ulcers caused by impaired arterial blood flow, the lower limb eruptions, similar to those on the upper limbs, presented as plaques with central keratotic plugs. Furthermore, while the keratotic plugs became scabbed and gradually disappeared, the rash healed without developing into ulcers or erosions during treatment, as observed in Figure 6. This healing process differed from that of typical arterial ulcers, leading us to consider the skin lesions on the lower extremity as being related to ARPD.

Regarding patients with DM, ARPD is considered to potentially arise from microcirculatory disturbances due to DM, where degeneration of collagen fibers in the superficial dermis or the accumulation of glucose oxidation products may trigger inflammation [6,15,16]. In this case, the resistant ARPD-related eruptions on the upper limbs that had shown no response to topical corticosteroids for over 2 years dramatically improved 2 months after initiating DM treatment and administering alprostadil. Although arterial and venous blood flow in the upper limbs was not evaluated by contrast-enhanced CT, the absence of swelling or erythema indicative of thrombosis suggested that blood flow impairment in the upper limbs may be milder compared to the lower limbs. Therefore, it is possible that the regression of the rash on the upper limbs was attributed to improvements in microangiopathy following DM treatment and alprostadil administration. While cases of ARPD improvement following DM treatment are rare [40], there is a possibility that longstanding, treatment-resistant ARPD may respond abruptly to DM management. Therefore, dermatologists should also consider the course of DM treatment when managing ARPD.

The eruptions on the lower extremity, believed to be related to ARPD, did not improve with DM treatment or alprostadil administration; however, they showed a tendency to improve as the blood flow to the lower limbs was enhanced through catheter treatment in vessels. Kuo et al. reported that the pathophysiological impact of atherosclerosis extends into the microcirculation distal to the atherosclerotic lesion [41]. Thus, ischemia due to PAD may induce microangiopathy in distal tissues, potentially leading to the development of ARPD-related eruptions similar to those caused by DM.

In this case, the pronounced blood flow impairment due to PAD might hinder improvement in the lower limb lesions with DM treatment and alprostadil administration alone. However, the addition of endovascular catheter therapy for PAD improved arterial and peripheral microvascular blood flow, which may have contributed to the resolution of the lower limb lesions. Additionally, the eruptions closer to the central area, where the blood flow was more readily restored, improved faster than those in the distal areas, supporting this hypothesis. To date, the relationship between ARPD occurrence and compromised blood flow in major vessels has not been discussed. However, the high incidence of vascular disorders, such as arterial hypertension or chronic venous insufficiency, among underlying diseases in ARPD patients [4], as well as our case’s treatment response, strongly suggests a connection between blood flow impairment in major vessels and ARPD onset. Moreover, both DM and PAD are known to influence peripheral microvascular circulation [6,15,16,40], with diabetes increasing the risk of PAD and PAD, in turn exacerbating the risk of diabetes [42]. These interactions underscore the possibility that ARPD, closely associated with diabetes, may also be linked to blood flow impairment in large vessels such as those affected by PAD.

While a skin biopsy of the lesions on the lower extremity could not be performed, the morphology and healing process of the eruptions suggest that they were related to ARPD rather than arterial ulcers. Although the strong connection between ARPD and DM-induced microangiopathy has already been studied, the potential link between ARPD and blood flow impairment in major vessels, as observed in our case, has not been adequately explored. Therefore, further accumulation of cases and research is needed to investigate this association.

## Figures and Tables

**Figure 1 medicina-61-00036-f001:**
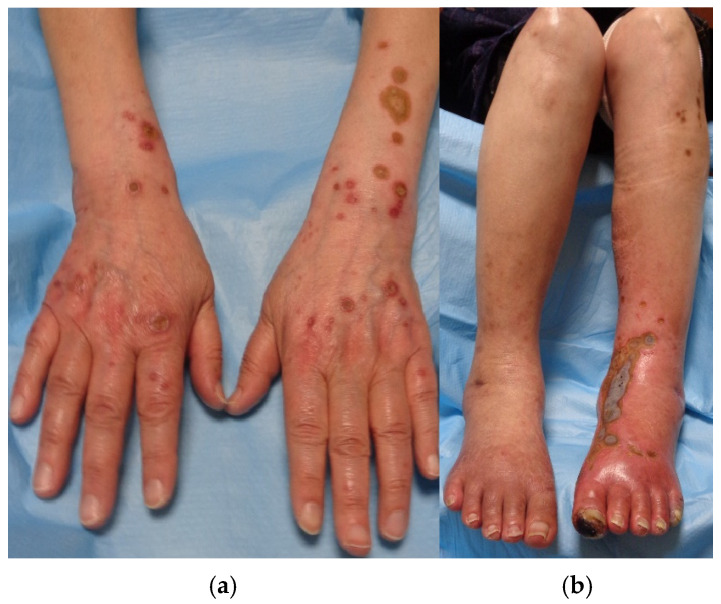
(**a**,**b**): Several papules and nodules on upper limbs, ranging from rice to bean size, accompanied by sebaceous plugs in the center (**a**). Swelling in the lower left leg, a black scab on the left index finger, and similar papules and nodules on the lower limbs (**b**).

**Figure 2 medicina-61-00036-f002:**
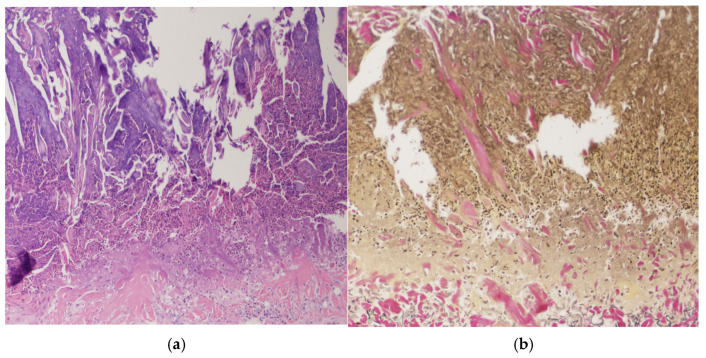
(**a**,**b**): Histopathological examination from upper extremity lesions revealed a cup-shaped defect, which contained keratin with parakeratosis, inflammatory cells, and necrotic material ((**a**): Hematoxylin and eosin stain ×40). Collagen fibers, vertically extruded from the superficial dermis, were also observed ((**b**): Elastica van Gieson staining ×100).

**Figure 3 medicina-61-00036-f003:**
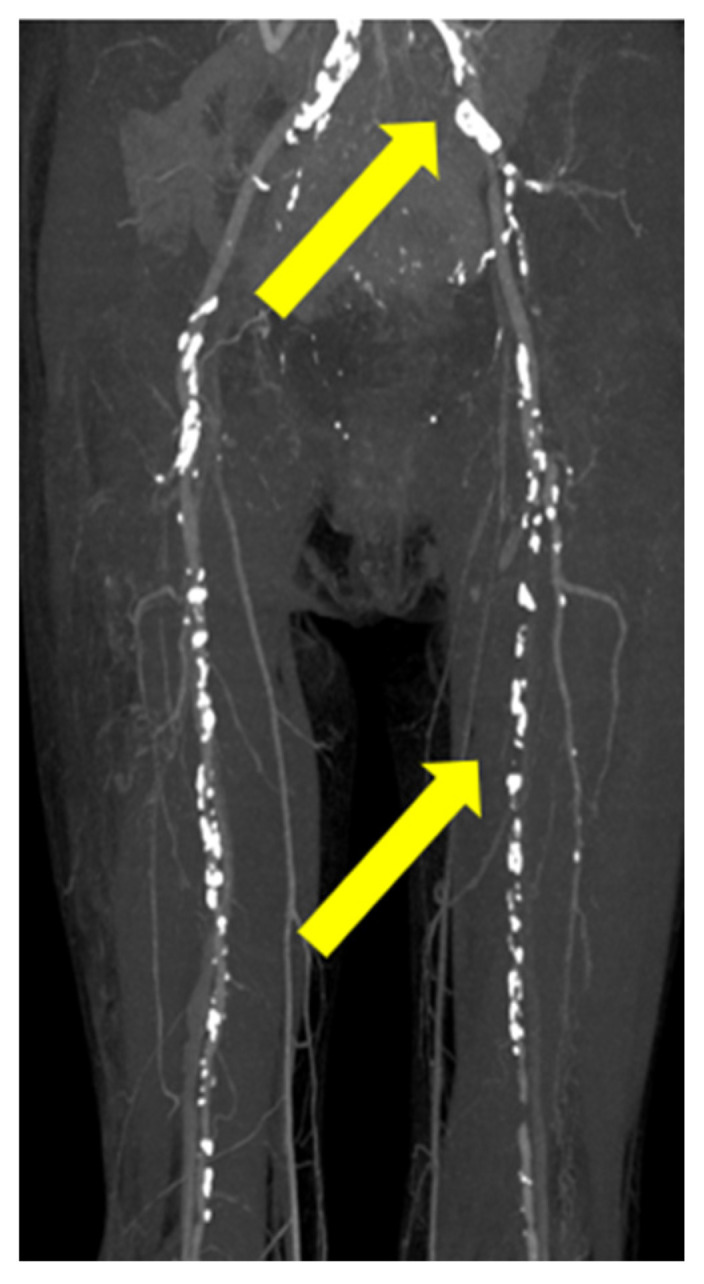
Contrast computed tomography revealed arterial narrowing and ischemia of the left common iliac artery and superficial femoral artery in the left lower limb (yellow arrow).

**Figure 4 medicina-61-00036-f004:**
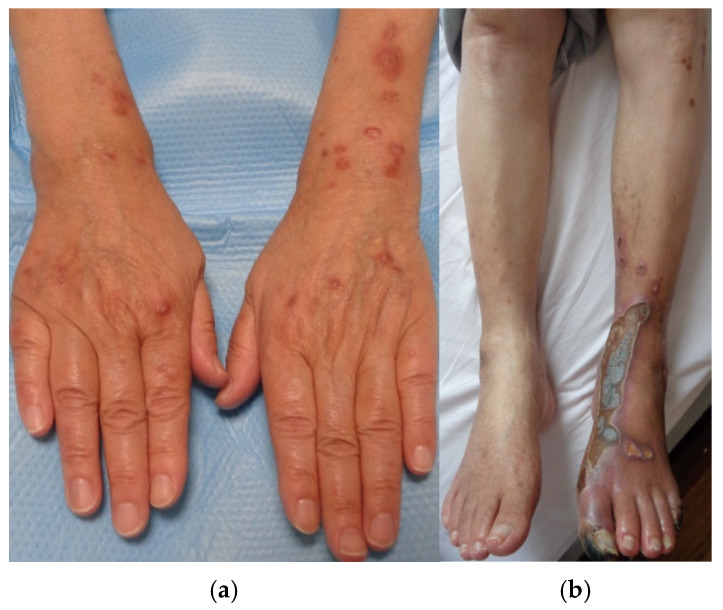
(**a**,**b**): Papules and nodules on the upper limbs remarkably improved after 2 months of diabetic treatment, whereas lesions on the lower limb remained unchanged.

**Figure 5 medicina-61-00036-f005:**
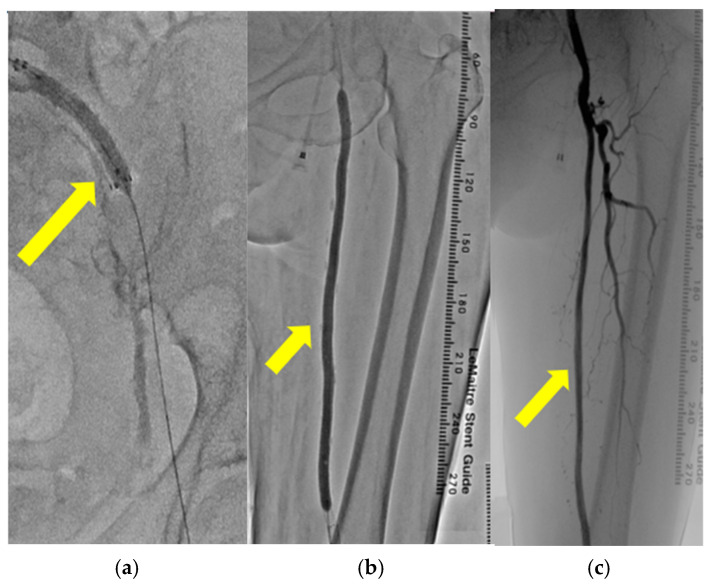
(**a**–**c**): Catheter angiography revealed the insertion of a stent into the left common iliac artery (yellow arrow) (**a**). Catheter angiography revealed balloon angioplasty performed on the left superficial femoral artery (yellow arrow) (**b**). Catheter angiography revealed enhanced blood flow in the left superficial femoral artery after endovascular therapy (yellow arrow) (**c**).

**Figure 6 medicina-61-00036-f006:**
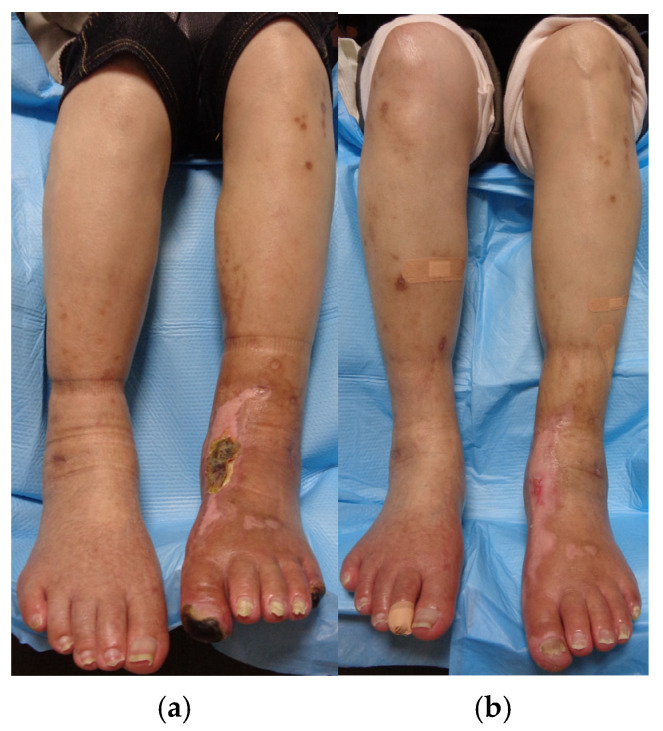
(**a**,**b**): The eruptions closer to the center of the lower leg showed complete regression 3 months post-endovascular therapy; however, the erythematous area with an internal large crust still left the crust inside (**a**). Eruptions on the lower limbs became almost flatly pigmented 6 months post-operation (**b**).

**Figure 7 medicina-61-00036-f007:**
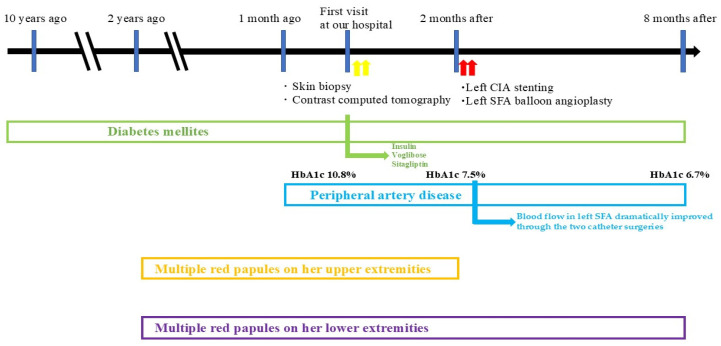
A table depicting the timeline and progression of the onset of DM, PAD, and skin lesions on the upper and lower limbs.

## Data Availability

The original contributions presented in this study are included in this article. Further inquiries can be directed to the corresponding author.

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
