# Peer review of "A Case of Acquired Reactive Perforating Dermatosis with Complete Resolution of Eruptions on Upper and Lower Limbs During the Treatment of Diabetes Mellitus and Peripheral Artery Disease"

_medicina, 2024, doi:10.3390/medicina61010036_

Round 1

Reviewer 1 Report

Comments and Suggestions for Authors

Mima et al. provide an interesting article regarding acquired reactive perforating dermatosis. 

I congratulate the authors for the well-written article and the high-quality images. I have a couple of suggestions: 

1. Case reports usually benefit from a figure displaying a timeline for each treatment and event. 

2. I would re-word the end of the discussion section. It is somewhat lengthy for an introduction and explains treatments of limited relevance for this case. The authors could more briefly mention them or even add this section to the discussion. 

3. If possible, I would revise a couple of odd references, especially those older than 20 years.

Author Response

Comment1:

Congratulate the authors for the well-written article and the high-quality images. I have a couple of suggestions: 

  1. Case reports usually benefit from a figure displaying a timeline for each treatment and event. 
  2. I would re-word the end of the discussion section. It is somewhat lengthy for an introduction and explains treatments of limited relevance for this case. The authors could more briefly mention them or even add this section to the discussion. 
  3. If possible, I would revise a couple of odd references, especially those older than 20 years.

Our reply to reviewer1:

Thank you for your detailed and thoughtful review of our case report in preparation for publication. Below are our responses and comments.

  1. Thank you for your feedback. The patient's medical history and treatment course were somewhat unclear, so in order to present this information more clearly, we have summarized the progression of DM, PAD, and the skin lesions on the upper and lower limbs in Figure 7.
  2. Thank you for your feedback. The section on ARPD treatment in the introduction part was quite lengthy, so we have revised it to be more concise without changing the main content (Line 89-105).
  3. Thank you for your feedback. Of course, we would have preferred to cite the most recent studies. However, as the concept of ARPD has remained largely unchanged over time, there are certain aspects where we had no choice but to refer to older references. We hope you can understand this limitation.

Reviewer 2 Report

Comments and Suggestions for Authors

The current manuscript presented a case with acquired reactive perforating dermatosis (ARPD) that exhibited a differential response to DM treatment and catheter surgical procedures in the lower and upper limbs. While the results are interesting and potentially informative to clinical practice, the current format of the manuscript is required to address the following issues:

1)The main concern is that the lesions in the lower limbs may differ from those in the upper limbs, indicating that different mechanisms might be involved. In Figure 3, a contrast CT scan showed narrowing and ischemia of the arteries in the lower extremities. Were similar findings observed in the upper limbs?

2)A better chronological description of the medical history is needed, including when the patient developed diabetes mellitus, vascular arterial narrowing, ischemia, and when the lesions on the upper and lower limbs began to appear.

3)Since a biopsy was not performed on the lower extremity lesions, it is necessary to establish an appropriate differential diagnosis including a few other skin conditions.

4)Did the patient experience itching or pain from the rashes on their upper or lower limbs? 

5)After the vascular surgical procedures, the lesions on the lower limbs show significant improvement. Are any other medications administered during and after the vascular surgical procedures?

6I would assume the diagnosis is Type 2 Diabetes. The patient was treated with insulin, voglibose, sitagliptin, and alprostadil. A more detailed description of these treatments is needed. How did the blood glucose and hemoglobin A1c levels change after the treatments?

7) Kidney failure could be the cause of ARPD. Did the patient have kidney issues when ARPD was diagnosed? 

Author Response

Comment2:

The current manuscript presented a case with acquired reactive perforating dermatosis (ARPD) that exhibited a differential response to DM treatment and catheter surgical procedures in the lower and upper limbs. While the results are interesting and potentially informative to clinical practice, the current format of the manuscript is required to address the following issues:

1)The main concern is that the lesions in the lower limbs may differ from those in the upper limbs, indicating that different mechanisms might be involved. In Figure 3, a contrast CT scan showed narrowing and ischemia of the arteries in the lower extremities. Were similar findings observed in the upper limbs?

2)A better chronological description of the medical history is needed, including when the patient developed diabetes mellitus, vascular arterial narrowing, ischemia, and when the lesions on the upper and lower limbs began to appear.

3)Since a biopsy was not performed on the lower extremity lesions, it is necessary to establish an appropriate differential diagnosis including a few other skin conditions.

4)Did the patient experience itching or pain from the rashes on their upper or lower limbs? 

5)After the vascular surgical procedures, the lesions on the lower limbs show significant improvement. Are any other medications administered during and after the vascular surgical procedures?

6) would assume the diagnosis is Type 2 Diabetes. The patient was treated with insulin, voglibose, sitagliptin, and alprostadil. A more detailed description of these treatments is needed. How did the blood glucose and hemoglobin A1c levels change after the treatments?

7) Kidney failure could be the cause of ARPD. Did the patient have kidney issues when ARPD was diagnosed?

Our reply to reviewer2:

Thank you for your detailed and thoughtful review of our case report in preparation for publication. Below are our responses and comments.

  1. Thank you for your feedback. The contrast-enhanced CT was performed from the neck to the lower limbs; however, it was not conducted for the upper limbs. There were no signs of swelling, erythema, or complaints of pain in the upper limbs that would suggest the presence of arterial thrombosis. Therefore, it remains unclear whether there was any reduction in blood flow due to arterial stenosis in the upper limbs. However, we believe that there were no significant thrombosis or stenosis in the lower limb arteries, as seen in the lower extremities. It is possible that the blood flow in the upper limbs improved with diabetes treatment and the administration of alprostadil. We have added this explanation to the Discussion section (Line 232-237).

  1. Thank you for your feedback. DM was diagnosed more than 10 years ago, but as the abnormalities were mild, the patient was monitored without active intervention. The skin lesions on both the upper and lower limbs appeared two years ago, while the symptoms of PAD developed one month ago. To present the progression of these conditions in a clear and detailed manner, we have summarized the timeline in Figure 7 as a graph.

  1. Thank you for your feedback. We have added the details regarding differential diagnoses and diagnostic considerations to the discussion section (Line 210-218).

  1. Thank you for your feedback. The skin lesions on the extremities were associated with itching. While the patient experienced pain due to swelling in the lower limbs, there were no complaints of pain specifically related to the lesions. This information regarding the itching and pain associated with lower limb swelling has been added to the case presentation (Line 120,121).

  1. Thank you for your feedback. During the endovascular procedure, anesthetic agents for pain relief and sedation, nitroglycerin as a vasodilator, iodinated contrast agents for vascular imaging, and intravenous antibiotics for infection prevention were used. Postoperatively, in addition to oral antiplatelet and anticoagulant medications, vasodilators such as alprostadil were administered. Therefore, the main focus of the treatment was to improve blood flow by promoting vasodilation and preventing thrombosis.

  1. Thank you for your feedback. At the initial consultation, the patient was not receiving any treatment for diabetes. Considering the need to control postprandial hyperglycemia, promote insulin secretion, and the likelihood of frequent contrast-enhanced examinations in the future, oral diabetes medications were selected. Specifically, sitagliptin, a DPP-4 inhibitor, and voglibose, an α-glucosidase inhibitor, were prescribed. Additionally, due to the significantly elevated blood glucose levels, around 400 mg/dL, insulin injections were initiated. Over the course of treatment, HbA1c levels showed a progressive improvement. These details have been added to the case presentation section (Line 159-164, 192-193).

  1. Thank you for your feedback. Throughout the treatment course, including from the initial visit, there was never an elevation in the creatinine levels. Therefore, we believe that the renal dysfunction did not have an impact on the development of skin rash. This information regarding normal renal function has been added to the case presentation (Line 150).

Round 2

Reviewer 2 Report

Comments and Suggestions for Authors

The case is definitely interesting to me and the issues raised in the previous review sessions were addressed. I am not concerned about the results in which the therapeutic responses of the upper and lower limbs are different. I am concerned that lesions in the upper and lower limbs are different, because the diagnosis of ARPD is depedent on histophatology. In addtion,  the antgiography was not performed in the upper extremeties. I would tone down the argument that the same type of skin lesion respond differentially to treatment.  Otherwise I am OK with the revised version. 

Author Response

Comment:

The case is definitely interesting to me and the issues raised in the previous review sessions were addressed. I am not concerned about the results in which the therapeutic responses of the upper and lower limbs are different. I am concerned that lesions in the upper and lower limbs are different, because the diagnosis of ARPD is depedent on histophatology. In addtion,  the antgiography was not performed in the upper extremeties. I would tone down the argument that the same type of skin lesion respond differentially to treatment.  Otherwise I am OK with the revised version. 

Our reply:

Thank you very much for your thorough and considerate review.

As you rightly pointed out, the expression suggesting that the lesions followed "different courses" might create a misunderstanding, implying an inconsistency in the treatment course for the same ARPD lesions. In response, we have removed phrases such as "followed different courses" and "resistant to DM treatment and alprostadil administration."

Instead, we revised the text to emphasize that both DM and PAD are interrelated conditions (reference 46 has been added), and that the improvement in both upper and lower limb lesions was ultimately achieved through the alleviation of peripheral microvascular disturbances via DM treatment, alprostadil administration, and endovascular catheter therapy. By framing the treatment as addressing a shared underlying mechanism in the improvement of peripheral microvascular circulation, we have clarified that the outcomes for both upper and lower limb lesions were consistent and aligned.

These revisions have been highlighted in red text in lines 210-212, 250-253, and 261-265.

Additionally, minor adjustments have been made to the title and abstract to align with these changes.

We would be grateful if you could review the revised manuscript at your convenience.